# Effect of Stocking Density during CO_2_ Stunning of Pigs on Induction Time and Activity Level Measured Using AI

**DOI:** 10.3390/ani14131953

**Published:** 2024-07-02

**Authors:** Rikke Bonnichsen, Claus Hansen, Jon Raunkjær Søndergaard, Dorte Lene Schrøder-Petersen

**Affiliations:** Department of Food and Production, Danish Technological Institute, Gregersensvej, DK-2630 Taastrup, Denmark; clha@teknologisk.dk (C.H.); jrso@teknologisk.dk (J.R.S.); dsc@teknologisk.dk (D.L.S.-P.)

**Keywords:** pig, stunning, CO_2_, loss of posture, animal welfare, artificial intelligence

## Abstract

**Simple Summary:**

The most extensively used method in Europe for stunning pigs before slaughter is exposure to carbon dioxide (CO_2_) in high concentrations (>80%). However, this method is controversial because it can result in variable degrees of aversive reactions during gas exposure. The variation in reaction among pigs is believed to be affected by different stressful conditions before or during the stunning phase. The stocking density in the gondola may be one factor influencing how stunning proceeds. The aim of this study was to investigate the effect of stocking density on the stunning progress of CO_2_ under commercial conditions. Both pigs’ reaction to the gas (measured activity level using artificial intelligence (AI)) and induction time (time to loss of posture) were reduced by reducing the number of pigs in the gondola. The result indicates that pig welfare during CO_2_ stunning can be improved by reducing the stocking density. The result also leads to a point of attention to the ongoing initiatives to find alternatives to CO_2_. When comparing different gas methods, the number of pigs can be a crucial factor in making the right conclusions. Additionally, the results show that vision-based activity measures could be a useful tool to evaluate stunning.

**Abstract:**

During the CO_2_ stunning of pigs, a variation in their reaction to the gas and the duration of the induction period is observed. The stunning process can be affected by several conditions, such as stressful events and previous experiences, but the stocking density in the gondola may also have an impact. The objective was to investigate the effect of stocking density on the stunning process under commercial conditions. To quantify the pigs’ reactions under industrial settings with a stocking density of up to eight pigs pr. Gondola (3.91 m^2^), the activity level was measured using an AI solution. Compared with a simulation of the expected induction period, a significantly longer induction period was found in gondolas containing seven and eight pigs (*p* < 0.001) but not when the gondolas contained three or four pigs. Both high and mean activity levels were significantly higher when stocking density was increased from three or four pigs to seven or eight pigs. The stunning process was thus negatively affected when increasing the stocking density. More knowledge is needed to explain this effect and to make statements on optimal stocking density. The measured activity levels may be a useful tool for obtaining information under commercial conditions and for documenting animal welfare.

## 1. Introduction

The practice of using a controlled atmosphere with CO_2_ to stun pigs prior to slaughtering is the most widely applied method in commercial settings in Europe. Currently, the only other commercially available alternative is electrical stunning. The principle when using CO_2_ is that the pigs are exposed to the gas at a high concentration until loss of consciousness [1]. Using CO_2_ offers several advantages. It permits the management of pigs in groups with minimal human intervention, thereby reducing stress. Moreover, it poses a low risk of errors by operators and ensures a dependable level of stunning effectiveness; when given sufficient exposure time, there is minimal chance of animals regaining consciousness [2,3]. Nonetheless, the method causes concerns regarding animal welfare because of the associated aversive behavior shown during exposure to the gas [4,5,6,7]. The reactions shown before the pig loses its posture (LOP) and thereby loses the ability to remain in a standing or sitting posture can be characterized as a set of conscious behavior patterns such as, e.g., excessive breathing and escape attempts. When the pig loses its posture, it indicates that the pig is entering into a state of very low awareness. Though there is still no consensus about at which exact point in the process the pig loses consciousness [2], most of the vigorous movement that can be seen primarily after LOP is caused by involuntary muscle contractions (excitation) due to the inhibition of some neurotransmitters, and activation of others, which causes hyperpolarization and depolarization, respectively, in the brain [8,9].

At the EU level, effort has been put into finding alternative methods for stunning pigs to provide a calm stunning method without an aversive response from the pigs [1,10]. In addition, work is performed to optimize the current method, including making different guidelines of best practices for the pre-slaughter handling of pigs [11].

In different experiments, a large variation is seen in the way pigs respond to CO_2_ stunning and in the duration of the induction time [1]. The reactions of pigs to CO_2_ have shown to differ, both in latency to loss of posture and in the degree of aversive behavior, depending on pig genetics, age, previous experience (e.g., type of housing systems at the farm), stress levels and handling procedures prior to stunning [12,13,14]. In addition, in humans, it is known that pre-operative anxiety and stress can lead to prolonged induction time and a requirement to increase the total dosage of anesthesia [15,16,17]. There is thus considerable evidence to support the hypothesis that the stress level of the pigs can influence the stunning process. Various factors can influence the amount of discomfort and stress experienced by the pigs in abattoir settings and prior to stunning. On the day of slaughter, the pigs are loaded, transported, unloaded, moved to a lairage, housed for a short period, and finally moved to the stunner [18]. Thus, they undergo several changes and are handled numerous times. A multitude of factors affecting the stress level of the pigs can, therefore, potentially be adjusted to facilitate a calmer environment for the animals and improve the stunning procedure. It is consequently a complex affair to create the optimal framework for the stunning of pigs.

There is a strong need for more knowledge about the effect of different potential stressors, such as prior handling to stunning and stocking density in gondolas, to be able to optimize the use of CO_2_ stunning. There is also a need for more focus on how alternative gas methods affect the pigs when used under varying conditions in the production environment and whether they will be able to significantly reduce or remove the aversive behavior shown by pigs when initially exposed to a gas.

Measuring welfare during stunning with CO_2_ (or another gas) under commercial conditions is challenging. The stocking density makes counting, recording, and interpretation of animal behavior difficult. For that reason, the experimental setups for testing alternatives (different gas types) to CO_2_ are often performed at a low stocking density or even on single animals [19,20]. However, some studies indicate that the stocking density influences the way the pigs respond to the gas [13,21], and this can thus become a critical factor for these experiments since the purpose is finding methods suitable under highly productive conditions. When trying to find alternative gas-stunning methods or when trying to improve the existing method, it is crucial that the comparison is made objectively and in a repeatable manner. Furthermore, the benefits of a given method discovered under experimental conditions must be the same under commercial conditions. Given the complexity and multifactored dependency, it could be a beneficial approach to use AI measures to reinforce the possibility of analyzing or interpreting large datasets [22]. The purpose of this study was to investigate the effect of stocking density on induction time and the way pigs react in the initial stunning phase during CO_2_ stunning of slaughter pigs under commercial conditions. To accomplish this, a vision system was developed to measure the activity level during the stunning phase using AI. Using this approach provides a way to assess the reaction of the pigs in the gondolas under difficult production conditions and thereby gives a uniform and reliable indication of the welfare during induction.

## 2. Materials and Methods

### 2.1. Animals, Lairage, Driving and Stunning Procedure

Data were collected on 10–11 November 2022 at a Danish commercial slaughterhouse, following the industrial standards for Danish slaughterhouses when handling live animals. The slaughterhouse in this study handled approx. 45,000 pigs per week. Upon their arrival, pigs were kept in pens with 15 pigs per pen, each pig with an approx. 115 kg live weight. The pigs were a mix of females, castrates, and entire males. Pigs were driven manually from the unloading area to the lairage. From the lairage to the stunner, the pigs were driven using automated driveway systems with mechanical push-hoist gates. The stunning system was a Marel Backloader 7 (3.91 m^2^) Gondola RelaX-XL (Garðabær, Iceland) with seven gondolas. The stunning system applied CO_2_ in high concentrations during stunning, when the gondolas descended into a CO_2_ concentration of at least 90 percent at the bottom position. The duration of gas exposure was at least 165 s. Pigs were driven towards the stunner in groups of 15 pigs. The pigs were divided into smaller groups in the driveway close to the area before entering the stunner. The pigs were stunned in group sizes of 3 to 8 pigs according to the production capacity after stunning. When slaughtering with high capacity, the 15 pigs in a group were divided into two gondolas of 7 and 8 pigs, and when slaughtering with lower capacity, the 15 pigs were divided into three gondolas of 3, 3, and 4 pigs, respectively. The slaughterhouse mainly produced at high capacity from morning to early afternoon, meaning that the animals were typically stunned in groups of 7 or 8 in each gondola. However, exceptions could be seen in connection with employee breaks or in case of a minor production stop. From late afternoon until early evening, the pigs were typically stunned in groups of 3–4 animals at a time. Thus, most gondolas contained 3, 4, 7, or 8 pigs, and only gondolas with 3, 4, 7, or 8 pigs were included in the study.

### 2.2. Recording of Video during Stunning

An overview of each gondola during stunning was recorded using a camera (GoPro HERO10, San Mateo, CA, USA) mounted with an external battery in a custom-made waterproof casing. The camera was mounted on the top rails in the center of the gondola with a top-down view. The camera was set to 24 frames per second with a resolution corresponding to HD set to “SuperView basic”, which corresponds to a field of view of approx. 113 degrees horizontal and 86 degrees vertical.

### 2.3. Analysis of Video Recordings of Animal Behaviour

Each video of a stunning procedure was evaluated manually and comprised a total of 203 recorded stunning procedures (gondolas). For each gondola, the time for the pigs’ first response to the stunning gas and the time of loss of posture (LOP) for the first and the last pig were determined (Table 1). Loss of posture was registered since it is the earliest visible sign possible to register in the given conditions to indicate the onset of unconsciousness during exposure to CO_2_ [1].

For a subset of randomly sampled gondolas with 3 or 4 pigs per gondola, the time to loss of posture for each individual pig in the gondola was recorded. With 3–4 pigs in the gondola, it was possible to determine the pigs’ individual LOP under conditions in which the pigs exhibited a limited physical impact on each other. These recordings comprised a total of 130 pigs and were recorded to establish a baseline dataset in this study for time to loss of posture (LOP). This baseline dataset was used to generate a simulated expected time-to-last loss as the basis for the comparison between gondolas with respect to time-to-last loss of posture.

For every other gondola, the time was determined until the first pig experienced LOP and until the last pig in the gondola experienced LOP.

Pigs that were still standing in an upright position sometimes covered the view of other pigs, making the first LOP subject to some uncertainty. In these cases, the LOP of the first pig was registered for the first sufficiently visible pig. Even with this uncertainty, registrations provide a timestamp for when the first pigs in the group lost their ability to remain standing. Recordings of the first response and loss of posture were made manually according to strict criteria (Table 1). The assessments were conducted by two observers. To ensure consistent assessments, an inter-observer calibration was performed before the assessments were made. During the period when the assessments were conducted, ongoing sample checks (recordings from approximately 30 gondolas in total) were made on the same sequences to ensure the maintenance of consistent assessments.

For each gondola, the level of activity per time unit was recorded using an artificial intelligence model to evaluate the relative change in pigs’ position between video frames (still pictures). The activity level was calculated from segmented frames isolating only pixels containing pigs.

A prerequisite for the method was to only look at movement in pixels containing pigs. This was because the average movement of each frame, hence pixels not containing pigs, would cause the average movement of the frame to go down, as there was no movement in these pixels. Therefore, the background was segmented out (Figure 1A).

A deep learning segmentation model with U-net architecture and a ResNet-18 backbone was used. The model was trained on images extracted from a variety of sedation videos to obtain a diverse dataset. It was not possible to segment out each individual pig, as the blurring, caused by the movement of the pigs, made it too difficult. Therefore, all the pigs were segmented as a whole, and the background was removed.

The frame-to-frame activity was calculated using a deep learning model with a Recurrent All-Pairs Field Transform (RAFT) architecture trained on the KITTI dataset [23]. This particular model was chosen as it has strong accuracy and strong cross-dataset generalization [23]. The RAFT model took in two frames at a time and calculated the optical flow, i.e., the velocity of movement in each pixel, by comparing the two frames (Figure 1B). Velocities for all pixels in each frame were obtained by feeding the model the current and the previous frame. For each frame, the average speed of all pixels containing pigs was calculated, thus providing a measurement of the level of activity for each frame.

The level of activity in each gondola was measured as the average activity level per frame from the first response until the time for loss of posture for the first and the last pig in each gondola.

The amount of high activity in each gondola was calculated based on the activity measurements in all frames from the first response to the time when the first and the last pig experienced a loss of posture. High activity was defined as the activity level above the median of all video frames in all gondolas from the time of the first response until the last pig experienced loss of posture. The amount of high activity and mean activity was calculated from the first response until both the first and last pig experienced loss of posture.

The method was validated on a small scale by testing the output against video observations of 30 gondolas. The observations were made in gondolas, for which the recording of activity, measured by the AI model, spread over numerically high, intermediate, and low activity levels. A subjective assessment was made of the recordings using a calibrated categorization of severity grading as mild, moderate, significant, or intense. The video observations were made by two trained observers. The activity level was divided into three levels of severity. The measured severity of activity was coherent with the grading of severity made by the observers.

### 2.4. Statistical Analysis

Statistical analysis was performed in R statistical software (version 4.3.1), and all significance levels were set to 0.05.

The effect of stocking density on the stunning process was evaluated using the parameter loss of posture and the activity level. The effect parameter “Loss of posture” was determined as the time from the first response until the last pig in a gondola experienced loss of posture (LLOP). A direct comparison of LLOP between gondolas with a varying number of pigs was not possible when the effect parameter was the time for the last observed pig experiencing “loss of posture” because the probability of a longer time to LLOP increases when the number of pigs in a gondola increases. Therefore, a simulated expected time to LLOP for gondolas with 7 or 8 pigs was calculated using the baseline dataset comprising 130 observations of time to loss of posture from gondolas with 3 or 4 pigs. The dataset of simulated expected time to last loss of posture (simLLOP) for gondolas with 7 or 8 pigs was generated by resampling from the baseline dataset and generating an equal amount of data points as the observations for last loss of posture in the group of 7 and 8 pigs in the gondola, respectively. The difference between LLOP and simLLOP was analyzed for gondolas with 7 and 8 pigs using the Mann–Whitney U Test. A non-parametric test was chosen due to the lack of variance homogeneity between groups.

The difference in activity between gondolas with a varying number of pigs was evaluated using the calculated frame-to-frame pixel velocity. The effect on mean activity and sum of high activity was analyzed for the duration from the time of the first response until the point when the first, as well as the last pig, experienced loss of posture. Differences between groups were evaluated using the Mann–Whitney U test, and Bonferroni corrected the *p*-value. A non-parametric test was chosen due to the lack of variance homogeneity between groups.

## 3. Results

### 3.1. Time from First Response to Loss of Posture

The effect of stocking density in the gondola on the induction time was estimated using the time to loss of posture and was analyzed by comparing the observed time to the last pig that experienced loss of posture in gondolas with seven and eight pigs to a simulation of the expected time to the last pig experienced loss of posture.

During the experimental days, video recordings were acquired from a total of 10, 57, 59, and 77 gondolas with 3, 4, 7, and 8 pigs, respectively (a total of 1287 pigs). The number of gondolas for each group varied as a consequence of performing the study under the given commercial conditions. The individual time to loss of posture was recorded for 130 pigs in gondolas with three or four pigs. The mean time to loss of posture for the individually recorded loss of posture was 13.4 s (SD = 3.8 s). Based on the individually recorded loss of posture, a simulated time for the last pig experiencing loss of posture for gondolas was calculated for group sizes of seven and eight pigs (simLLOP). The time to the last pig experienced loss of posture (LLOP) was significantly higher than the simLLOP for both gondolas, with seven (*p* < 0.001) and eight pigs (*p* < 0.001) (Figure 2).

### 3.2. Mean Activity Level during Stunning

The effect of stocking density on the pigs’ behavioral reaction to CO_2_ was estimated using AI measurement of the activity. The mean frame-to-frame activity was calculated for each gondola from the time from the first response until the first pig in a gondola experienced loss of posture and until the time the last pig experienced loss of posture. The mean activity (Figure 3) was statistically higher in gondolas with higher stocking density. The effect was seen from the first response until the first pig experienced loss of posture, as well as until the last pig in the gondola experienced loss of posture (Table 2).

### 3.3. Levels of High Activity during Stunning

The effect of stocking density on the pigs’ reaction to CO_2_ was estimated by comparing a calculation of the sum of measured high activity during stunning for each gondola. The high activity level was defined as the frame-to-frame activity above the median activity for all video frames for all gondolas from the time of the first response until the last pig experienced loss of posture (Figure 4). There was no difference between the stocking density of three and four pigs, and all other comparison combinations were significantly different (Table 3).

## 4. Discussion

### 4.1. Criteria for Method Selection

There is an increasing focus on optimizing animal welfare during the stunning of pigs. Consequently, several studies have been carried out to investigate whether it is possible to optimize the existing method or use alternative gases or gas mixtures to stun pigs to provide a calmer process with less aversion to the gas [3,14,20].

To evaluate the pigs’ response to a given gas, the typical approach is to make an ethogram consisting of behavioral elements expressing negative welfare and the level of consciousness. Traditionally, the ethogram involves recording behavioral characteristics that can indicate discomfort, distress, or fear (e.g., walking, gasping, headshaking, vocalization, backing, mounting, jumping, crawling), drowsiness (sitting, loss of posture, lying down), and muscle contraction (opisthotonos, excitation) [2,13,14,24,25]. The commercial conditions, i.e., pigs being gas-stunned in a gondola at a high stocking density, make it challenging to create the best prerequisites for using an ethogram as a method to assess the stunning process. To make a full behavioral ethogram containing the different behavioral patterns that are shown in response to the stunning gas, it is essential that the behavior observed is clearly defined for consistency. The stocking density makes it problematic to study the stunning process on an individual level and to distinguish between the different behavioral patterns. Consequently, in the existing scientific literature, the experiments that have been carried out regarding the stunning of pigs are often performed on a smaller group of animals or even on single animals in set-ups that allow for a visual inspection [14,26].

However, this approach is problematic if the stocking density in the gondola per se has an impact on the stunning process. This makes it difficult to transfer the results from testing in a small-scale laboratory set-up to real-time production conditions. In this study, we aimed to test the importance of stocking density in a commercial setting, using a method developed to make an objective assessment of the pigs’ response to stunning gas under production conditions.

### 4.2. Method of Behavioural Assessment

To assess animal welfare, it is crucial to choose the period in which the animals may experience discomfort, i.e., the period from the pigs’ start to react to the gas until they lose consciousness. However, it is complex to distinguish exactly when the pigs are completely unconscious, as it is a gradual process, and there is a lack of consensus in the existing literature.

“Loss of posture” (LOP) was, in this study, defined as the point when the pig loses the ability to remain in a standing or sitting posture. LOP is used in this study, as in several other experimental studies, as an indicator of the transition to loss of consciousness [6]. It was possible to determine the pigs’ first reaction and LOP for the first (sufficiently visible) pig and the last pig by manual observations of the recording. Emphasis was therefore placed on finding the pigs’ first reaction to the gas and LOP and then on assessing the behavior in the time from the first reaction until LOP. This study then relates to the induction time (from the first response to LOP) and how the pigs react in this period.

The prerequisite for selecting the method to evaluate the pigs’ response to the gas was that, in general, it is desirable to obtain a stunning during which the pigs rapidly lose consciousness without vigorous escape attempts [11]. I.e., obtaining a calm and fast induction of stunning. Given that the stunning process occurs in an unfamiliar and stressful situation, it is reasonable to assume that positive behavior will not appear in the gondola during the period in which pigs are exposed to the gas. This means that the degree of increased activity in this period is caused by increased negative responses. This was supported by the manual observations of the video recordings completed to register the first response and LOP. Even though different behavioral patterns could not be counted at the individual level at a high stocking density, the behavioral responses that appeared after the first response were in line with the behavioral patterns used in other studies to indicate negative responses (Section 4.1). This was also supported by the small-scale validation made. Consequently, the method of summing the activity level of the pigs is assessed to be operational in determining whether negative responses from the pigs escalate or diminish.

There are some obvious limitations to the method since not all behavioral responses are detectable when using AI to sum the activity in the gondola. Consequently, there will be some nuances of the animals’ behavior that are not recorded with this method, and the individual aspect will be lost. One aspect is that the method does not relate to the animal’s vocalization or the frequency of gasping. These parameters, which are considered important in evaluating distress [3,27], are neither recorded nor part of the present study.

Since the method detects all movements, it is not able to distinguish between what is caused by uncontrolled muscle contraction/excitation and what is caused by conscious movements. For either approach (making manual registrations or measuring activity level), attempting to discern these types of motions will depend on the time at which the animal loses consciousness and whether the movements are observed before or after this point (indicated by LOP). Furthermore, using activity level measures does not differentiate between specific patterns–backing, jumping, turning, etc. and will not be able to count the frequency of the different patterns. The significance of this will depend on the purpose of the study. In general, it can be difficult to assess specific behavioral patterns according to severity. For example, it is not necessarily clear whether backing indicates a more stressful state of the pig than jumping.

Overall, there are some limitations to the method of measuring the activity (frame-to-frame pixel velocity). Even though the method is generic, it provides a relative scale that is not directly transferable to a different technical setup. However, it is a major benefit that the method provides an objective way to analyze a large amount of data under commercial conditions in a cost-efficient way that also allows for easy replication of the experiment by others. Furthermore, the method provides an opportunity to analyze activity patterns and gain information about the typical patterns and when patterns deviate.

Furthermore, the method has a great potential to achieve the general goal of a quick and calm induction of stunning as it continuously monitors the pig’s activity in the stunner, ensuring that the stunning takes place in a calm manner. For this, a threshold can be set to give an alarm when the activity reaches a certain level. Moreover, the general activity level can be used when testing and comparing different gases for stunning during high stocking densities or to evaluate if a given measure taken at a slaughterhouse has an impact on the stunning process.

### 4.3. Study Results

The results show that the stunning process is negatively affected by increasing the stocking density in the gondola.

When comparing the induction period, measured as the time from the first response to the last pig experienced loss of posture (LLOP), to a simulation of the expected time to LLOP, the time was significantly higher in gondolas with seven (*p* < 0.001) and eight pigs (*p* < 0.001). For gondolas with three or four pigs, no significant difference was found from the simulated expected time. This reveals a prolonged induction time when increasing the number of pigs in the gondola to seven or eight pigs.

The pigs’ response to the gas, assessed using AI measurements of the activity level, also increases with increasing number of pigs in the gondola.

In this study’s analysis, a differentiation was made by looking at the mean activity level and the amount of high activity level in the period from the first response until the LOP of the first pig, as well as in the period from the first response until the LOP of the last pig.

In the period from the first response until the LOP of the first pig, the activity level was only measured while all animals were standing up and were conscious to some degree. This means that the activity level (the pigs’ reactions) that occurs here was caused by and experienced by all pigs. In the period from the first response to the last LOP activity, the measures include a period when some animals were lying down and may have had uncontrolled muscle movements while others were still standing. Thus, this forms a picture of what was experienced by the last animals to lose consciousness. In both cases, the amount of high activity and mean activity was significantly higher when the stocking density was increased when comparing three or four pigs to seven or eight pigs. Thus, these results point in the same direction regardless of whether the period is until the first LOP or until the last LOP. This demonstrates that stocking density seems important for the process, both when all pigs are standing and until the last pig has lost posture.

This study does not explain why the stocking density has this influence on the stunning process, and there can be a multiple number of possible explanations.

For one, the premise of this study is that it is conducted under commercial conditions. This means that conditions throughout the chain prior to stunning can potentially affect the stunning process. Nonetheless, all pigs were subjected to the same procedure from arrival to slaughter, as the slaughterhouse operates according to standard procedures, including making continuous measurements of CO_2_ in the stunning system. Thus, there were no visible systematic conditions that could blur the conclusion. However, there could be a difference in the stress impact the pigs experienced just before stunning, which could contribute to the explanation. At the time when the animals were divided into several groups, the driving pace just before the stunner was slightly reduced when only 3–4 pigs were placed in each gondola. This can mean that the pressure just before stunning on the groups of 3–4 pigs may have been less than for groups of 7–8 pigs. However, this study was not set up to identify to which extent the conditions in the last minute before stunning could be a contributing factor to the increased activity during stunning.

A general factor addressed by [1] is the problems that can occur when exceeding the capacity of the stunning equipment in terms of the number of pigs loaded into the gondola. It is highlighted that overloading increases the risk of unnecessary excitement that may lead to an increased risk of bruising. In a group stunning situation, overloading the gondola may also lead to animals falling on top of each other, and consequently, compression of the chest of some animals potentially leads to inadequate exposure to the gas. However, there is no fixed limit for overloading the gondola, and the number of pigs in each gondola in this study did not exceed the guidelines for the equipment used.

Another possible explanation can be that it is generally more stressful for the pigs to be in a gondola if they are pressed close together. The instinct of the pig may influence the process since pigs are herd animals and warn each other of potential danger [28]. An increased number of pigs can resolve in increased alertness and response. The probability of escape behavior increases with more animals present because there are more pigs that can escalate alert behavior or anxiety through social facilitation.

Furthermore, more pigs in the gondola mean that the pigs push each other to a greater extent and thereby potentially influence other pigs’ process of “phasing in” to unconsciousness during the stunning.

Overall, the results indicate that there is a potential to achieve a faster and calmer stunning process when using CO_2_ in high concentrations by reducing the number of animals in the gondola. This study does not relate to whether a change in the stocking density achieved by increasing space instead of reducing the number of pigs will have the same effect. Overall, more knowledge is needed to explain the effect shown and to make recommendations for an optimal stocking density.

Another important aspect of these results is that they confirm that it is extremely important to be aware of the setup and the methods used to test the potential alternatives to CO_2_ stunning. It must be ensured that an alternative can lead to increased animal welfare when it is used in commercial settings. Little is known about the extent to which pigs’ reaction to other types of gas for stunning would be influenced in the same way by stocking density under commercial conditions. It could, however, influence the interpretation of results found under experimental setups with a small number of animals if that is the case.

The possibility of testing and documenting how animals react in the gondola directly by using an AI solution under commercial conditions has several advantages. By testing alternative stunning methods under commercial conditions, it will, to a greater extent, be possible to map the importance of all the other pre-slaughter factors that may be linked to the variation in how pigs respond to gas. Furthermore, continuous surveillance and the possibility of collecting a large amount of data will provide an opportunity to look for general patterns and relevant deviations from these patterns.

## 5. Conclusions

The number of pigs in the gondola during the CO_2_ stunning, comparing three or four pigs to seven or eight pigs, had an impact on the stunning process. Both the induction time and response, measured as activity level, increased with higher stocking density in the gondola.

This study illustrates that the stocking density under commercial conditions is an important factor that needs consideration when evaluating a stunning process. Analyzing the activity level using AI could be a useful tool to address the difficulty of evaluating pigs during commercial conditions at a high stocking density. Using AI provides a continuous objective scale that is easy to apply and has a temporal resolution, which behavioral ethograms have difficulty capturing.

## Figures and Tables

**Figure 1 animals-14-01953-f001:**
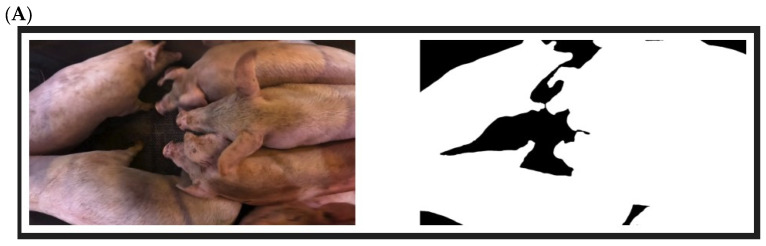
(**A**). Left: picture in the gondola used during stunning. Right: sequencing of pixels containing pigs to be able to determine the average speed of all pixels containing pigs’ movements. (**B**). Left: sequencing of pixels containing pigs in a gondola with 3 pigs. Right: a still picture illustrating, by vectors, the velocity and direction of movement in the pixels containing pigs.

**Figure 2 animals-14-01953-f002:**
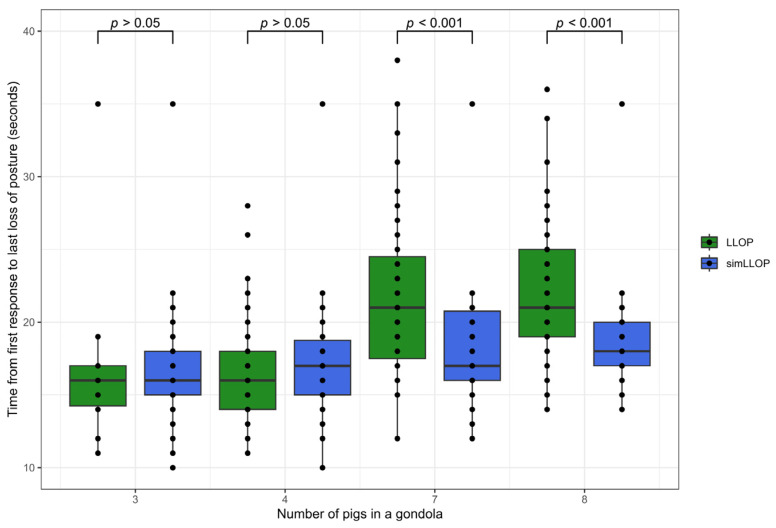
Box and whisker plot of simulated versus observed time from the first response to CO_2_ until the last pig in the gondola experienced loss of posture (simLLOP versus LLOP). Data are shown for gondolas with 3, 4, 7, and 8 pigs.

**Figure 3 animals-14-01953-f003:**
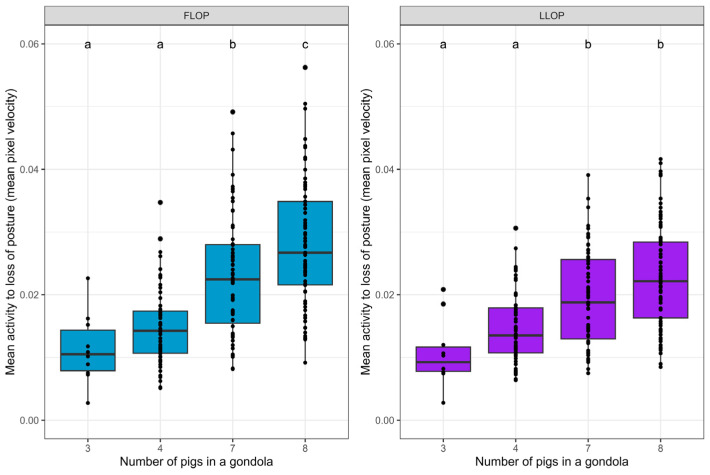
Box and whisker plot of the mean activity level for all pigs during stunning for group sizes of 3, 4, 7, and 8 pigs in each gondola. The mean activity measured as mean pixel velocity was calculated from the time of the first response until the first pig (FLOP, **left**) and the last pig (LLOP, **right**) in the gondola experienced a loss of posture. Letters (a, b, c) denote significant differences (*p* < 0.05). The *p*-values are seen in Table 2.

**Figure 4 animals-14-01953-f004:**
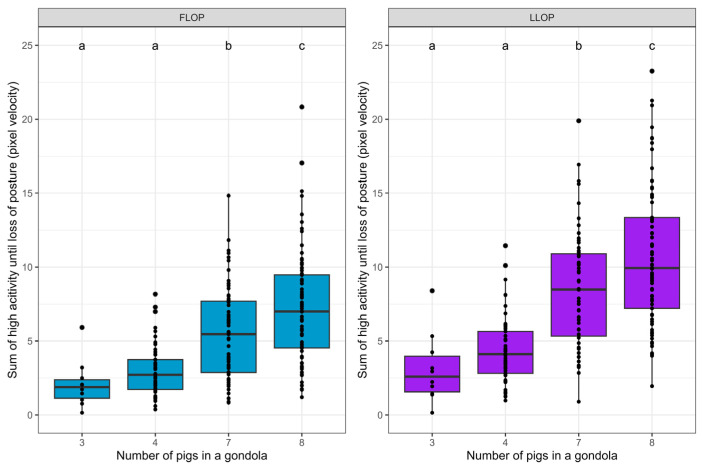
Box and whisker plot of the sum of high activity for pigs during stunning for group sizes of 3, 4, 7, and 8 pigs in each gondola. The sum of high activity was calculated from the time of the first response until the first pig (FLOP, **left**) and the last pig (LLOP, **right**) in the gondola experienced a loss of posture. Letters (a, b, c) denote significant differences (*p* < 0.05). The *p*-values are seen in Table 3.

**Table 1 animals-14-01953-t001:** Criteria for manual evaluation of time of first response to the anesthetic gas (CO_2_) and loss of posture for pigs in the gondola.

Criteria	Definition
First response	A response is recorded after the gondola has commenced the descent. At least one pig responds with alertness by either a sudden, typically small, movement (most pigs are not moving just before the descent) or raising the head with increased sniffing.
Loss of posture (LOP)	The pig either lies down and is no longer able to regain a standing or sitting posture, or the pig remains in a sitting posture against the wall or other pigs while displaying opisthotonos, indicating that the pig is not able to remain in a sitting posture, but is mechanically prevented from lying down.

**Table 2 animals-14-01953-t002:** Differences between groups of gondolas with different group sizes. Bonferroni corrected *p*-values represent differences in mean activity levels from the time of the first response until the first pig experienced loss of posture (FLOP) and until the last pig experienced loss of posture (LLOP) (data shown in Figure 3).

Group Size in Gondola	Group Size in Gondola	FLOP (*p*-Values)	LLOP (*p*-Values)
3	4	0.4115	0.2866
	7	0.0015	0.0052
	8	<0.0001	0.0002
4	7	<0.0001	0.0049
	8	<0.0001	<0.0001
7	8	0.0210	0.1090

**Table 3 animals-14-01953-t003:** Bonferroni corrected the *p*-value for differences between gondolas with different group sizes. *p*-values represent differences in the sum of high activity levels from the time of the first response until the first pig experienced loss of posture (FLOP) and until the last pig experienced loss of posture (LLOP).

Group Size in Gondola	Group Size in Gondola	FLOP (*p*-Values)	LLOP (*p*-Values)
3	4	0.5782	0.3513
	7	0.0036	0.0004
	8	<0.0001	<0.0001
4	7	<0.0001	<0.0001
	8	<0.0001	<0.0001
7	8	0.0274	0.0364

## Data Availability

Data obtained from the slaughterhouse for this study are not openly available due to privacy concerns. However, they can be made available upon request from the corresponding author. Please contact the corresponding author for access to data.

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
