# Peer review of "Effect of Stocking Density during CO2 Stunning of Pigs on Induction Time and Activity Level Measured Using AI"

_animals, 2024, doi:10.3390/ani14131953_

Round 1

Reviewer 1 Report

Comments and Suggestions for Authors

Author Response

Thank you very much for taking the time to review this manuscript. Please find the responses below and the corresponding revisions in track changes in the re-submitted file

Comment 1:
Selection of Gondolas based on the number of animals
Line 123-124: “Only gondolas with 3, 4, 7, or 8 pigs were included in the study”
Line 134: 203 gondolas were evaluated
It is not entirely clear to me whether the gondolas on the two study days contained exclusively 3, 4, 7, or 8 animals, or if gondolas with other numbers of animals were also processed, but only those with 3, 4, 7, or 8 animals were included in the study.
If the former is the case, I suggest revising the sentence in lines 123-124 to clarify that only these animal numbers were processed in the gondolas during the two study days.
If the latter is the case, could you provide the following details:
- The number of gondolas excluded from the analysis and the number of animals they contained
- The reason for choosing to analyze only the gondolas with 3, 4, 7, or 8 animals, and not those with other numbers of animals.

Response 1: Thank you for pointing this out. I agree that this could be clearer, and I added some additional text to describe this in section 2.1. Animals, lairage, driving and stunning procedure.

Comment 2
Line 124-135: “For each gondola, the time for the pigs´ first response to the stunning gas was determined”.
Was this assessment conducted by a single person or several people, and more importantly, by the same individual throughout? Please add this information if available.
Additionally, when I first read the article, it was unclear to me which criteria were used to assess the initial response to the gas. It may be beneficial to refer to Table 1 at this point in the manuscript.

Response 2: Thank you for pointing this out. Elaborations are made in text in line 160-164 and a reference to the table put in line 142.

Best regards Rikke

Reviewer 2 Report

Comments and Suggestions for Authors

This study has investigated the effects of stocking density on the carbon dioxide stunning process under commercial conditions by AI. The topic is interesting. The study found that pig welfare during carbon dioxide stunning can be improved by reducing stocking density. The experimental design is fine. The following revision could improve the quality of the paper.

How about the accuracy of data was generated from AI?

L16, what is the meaning of AI? The full name should with it when it was firstly appeared.

The conclusion sentence should be added in the Abstract section.

P value need to be written in italic.

This reviewer found that the authors did not cite any references in the methods and material section. Please add the references.

L121, what is the meaning of ‘e.g.’?

L205-206, the authors have set the P value for the difference. Are there any significant changes among the groups in the Figure 2 and 3? The authors need labeled the statistical analyzed results in the figures.

L288, Table 3, what is the meaning of ‘A’ and ‘B’? The authors need to explained them in the footnotes. Please check the similar issues throughout the paper.

L384, please add space before and after ‘<’ and checking the similar issue throughout the paper.

Author Response

Thank you very much for taking the time to review this manuscript. Please find the responses below and the corresponding revisions in track changes in the re-submitted file

Comment 1: How about the accuracy of data was generated from AI?

Response 1: For this comment I refer to the new cover letter.

Comment 2: L16, what is the meaning of AI? The full name should with it when it was firstly appeared.

Response 2: Thank you for highlighting this. It has been corrected in the updated version.

Comment 3: The conclusion sentence should be added in the Abstract section.

Response 3: For this comment I refer to the new cover letter.

Comment 4: P value need to be written in italic.

Response 4: Thank you for highlighting this. It has been corrected in the updated version.

Comment 5: This reviewer found that the authors did not cite any references in the methods and material section. Please add the references.

Response 5: Two references are added.

Comment 6: L121, what is the meaning of ‘e.g.’?

Response 6: Thank you for pointing this out. This is a typing error and is now deleted.  

Comment 7: L205-206, the authors have set the P value for the difference. Are there any significant changes among the groups in the Figure 2 and 3? The authors need labeled the statistical analyzed results in the figures.

Response 7: Fig. 2,3 and 4 are altered to make this clearer.

Comment 8: L288, Table 3, what is the meaning of ‘A’ and ‘B’? The authors need to explained them in the footnotes. Please check the similar issues throughout the paper.

Response 8: Table 2 and 3 are altered to make this clearer.

Comment 9: L384, please add space before and after ‘<’ and checking the similar issue throughout the paper.

Response 9: Thank you for highlighting this. It has been corrected in the updated version.

Best Regards Rikke

Reviewer 3 Report

Comments and Suggestions for Authors

The present study evaluate the effect of stocking density on the pig welfare measures during CO2 stunning under commercial conditions via using AI. The findings of the study would be applicable in refining the gaseous stunning protocols. The language is clear and easy to understand.

                    i.            Simple summary: Appropriate

                  ii.            Abstract: L 29: Please mention the average size of gondola; Approx or average Bodyweight of pigs also if measured? Also please detail the AI solution also

                iii.            L 29-31: I would appreciate if authors mention the induction time here

                iv.            Keywords: May remove slaughter as the proposed meaning conveyed by stunning

                  v.            Moreover please mention clearly induction of unconsciousness or loss of awareness etc.

                vi.            L85-95: provide sufficient background information and support for the hypothesis. It is recommended to add some points on how AI could be useful in achieving the desired objectives. It will improve the hypothesis.

Methodology is well explained and detailed; for its further improvement I have following suggestions as-

              vii.            L109: please mention the stocking density if available

            viii.            L117-118: please mention the duration of gaseous exposure

Author Response

Thank you very much for taking the time to review this manuscript. Please find the responses below and the corresponding revisions in track changes in the re-submitted file

Comment 1-5:  

i. Simple summary: Appropriate

ii. Abstract: L 29: Please mention the average size of gondola; Approx or average Bodyweight of pigs also if measured? Also please detail the AI solution also

iii. L 29-31: I would appreciate if authors mention the induction time here

iv. Keywords: May remove slaughter as the proposed meaning conveyed by stunning

v. Moreover please mention clearly induction of unconsciousness or loss of awareness etc.

Response 1-5: Additional Information on bodyweight, gondola size and details of the AI solution are provided in the materials and methods section in the edited version. The keywords are changes – “Slaughter” is removed, and “Loss of posture” is added. A sentence to elaborate on the registration of loss of posture as the earliest possible sign (to record manually) to indicate the onset of loss of consciousness has been added in line 141. In relation to the abstract, I refer to the new cover letter.

Comment 6:

vi. L85-95: provide sufficient background information and support for the hypothesis. It is recommended to add some points on how AI could be useful in achieving the desired objectives. It will improve the hypothesis.

Response 6: An additional line 95 is added to highlight that using AI measures could be beneficial for analyzing and interpreting these data.

Comment 7-8:              

vii. L109: please mention the stocking density if available

viii.  L117-118: please mention the duration of gaseous exposure

Response 7-8: Thank you for highlighting this. The size of the gondola and the period of gas exposure has been added in line 115-118.

Best Regards Rikke